# Transformational Steam Infusion Processing for Resilient and Sustainable Food Manufacturing Businesses

**DOI:** 10.3390/foods10081763

**Published:** 2021-07-30

**Authors:** Christopher Brooks, Mark Swainson, Ian Beauchamp, Isabel Campelos, Ruzaina Ishak, Wayne Martindale

**Affiliations:** 1OAL, A1 Parkway, Peterborough PE2 6YN, UK; chris.brooks@oalgroup.com (C.B.); ian.beauchamp@oalgroup.com (I.B.); 2National Centre for Food Manufacturing, University of Lincoln, Holbeach PE12 7PT, UK; mswainson@lincoln.ac.uk (M.S.); ICampelos@lincoln.ac.uk (I.C.); ruzaina_ishak@yahoo.com (R.I.)

**Keywords:** sustainability, consumers, steam infusion, food processing, steam infusion

## Abstract

Here we show how food and beverage manufacturers report more incisive sustainability and product fulfilment outcomes for their business enterprises when innovative processing technologies are used. The reported steam infusion technology heats food materials within a Vaction Pump device so that steam is directed into the food material within a much reduced volume, reducing the use of steam and processing time. This study reports how such technological interventions will enable supply chain stakeholders to demonstrate responsible consumption by connecting assessments for the reduction of greenhouse gas emissions with consumer-focused outcomes such as product quality. The technology reported in this research not only improves operational agility by improving processing speed, but also improves the responsiveness of factory production to changes in demand. Heating procedures are systemic processes in the food industry that can be used to pasteurize, achieve commercially viable shelf-life, and provide cleaning in place. The reported research defines how these technologies can reduce the carbon footprint of products, improve quality attributes, and lower operating costs across supply chains. They provide an important step in developing distributed manufacturing in the food system because the technologies reported here are modular and can be installed into existing operations. The specific technology can reduce energy consumption by 17.3% compared to basic direct steam heating, with a reduction of 277.8 processing hours and 8.7 tonnes GHG emissions per kettle production line each year. Food and beverage manufacturers are increasingly required to report across the sustainability, nutrition, and product quality outcomes of their business enterprises more incisively so that supply chain stakeholders can demonstrate responsible production and consumption. The steam infusion technologies assessed in this research enable alignment to the UN Sustainable Development Goals, specifically SDG12, Responsible Production and Consumption, using in situ data logging in factory trials for novel heating procedures used to process foods.

## 1. Introduction

The merits of integrating efficient processing into food manufacturing operations are fully realised when they result in the sustainable outputs of improved resource utilisation, reduced food waste, and reduced greenhouse gas (GHG) emissions [1]. The research reported in this article establishes how food engineering applications can be integrated with sustainability reporting and consumer quality outcomes. While the change-agent is the technology (the Vaction^TM^ Pump device, manufactured by Olympus Automation Ltd. (OAL), Peterborough, UK), it is the connection to GHG emission, process time, and product quality that become transformative and are the focus of this research. The dominant direct energy consuming processes in food manufacturing factories are often heating operations, and innovative Steam Infusion cooking is tested here to highlight transformational changes across food manufacturing practices. The Steam Infusion technologies in this study have been developed and tested with OAL (OAL, Peterborough, UK) in the UK so that the improvements in manufacturing resilience in the food system made by using this technology can be reported. The Steam Infusion technology and equipment developed by OAL is patented and trademarked as the Vaction^TM^ Pump (see, Steam Infusion, see https://steaminfusion.oalgroup.com/ (accessed on 21 April 2020), personal communication from OAL). An ideal benefit of advanced processing technologies is the potential to integrate them into existing operations and develop a framework for reporting the sustainability attributes associated with every product. This is of specific use in adopting Industry 4.0 approaches, where there is a requirement to re-engineer existing processes to provide safer and more efficient outcomes through distributed manufacturing frameworks. Improving resource utilisation across supply chains is reliant on manufacturers investing in processing research and development (R&D), and the capacity for individual companies to do this varies [2]. It is typically determined by the ability of food companies to relate technology to return on investment (ROI), which is assessed in terms of improved production capacity, revenue generation, and increased brand value [3]. 

While each of these ROI attributes are well understood, the financial barriers to investing in sustainable technologies are often only reduced during reactive situations determined by crises that identify low resilience [4]. The current COVID-19 crisis has exposed critical points where there is low resilience and has highlighted aspects of food supply that confer greater resilience. One supply chain action identified as important to change is the increased supply or flow of product in response to intensified changes in demand. If production can be changed quickly so that it is responsive or resilient to changes in demand, then the restriction of supply can be ameliorated at the system level. The importance of COVID-19 has been identified at food system levels and technologies that can provide safe and modularised solutions that will offer the greatest benefit to many manufacturers because they can distribute controlling actions at critical points in supply chains [5]. The sustainability attributes of food manufacturing have not been a priority in this reactive space for many consumers because price and quality will still dominate purchase choice. Direct cost and volume attributes also dominate the route from new product development (NPD) and the scaling-up into production through to retailed manufactured product. This is changing because the ability to connect consumer experience to sustainability values has become a priority for many foods businesses. Sustainability credentials will strengthen brand values systemically across supply chains, and the inference of improved product leadership and safety in the food and beverage industries will strengthen ROI outcomes in future. 

Advanced processing technologies must provide methods of effectively delivering sustainability attributes into production by reducing manufacturing time and energy consumption per product. The financial cost of restructuring and integrating new processing into existing operations will be dependent on the ability to utilise R&D capability in what is a highly fragmented industry where food and beverage accounts for 13.3% of the total EU-28 manufacturing sector with a turnover of EUR 945 billion. Over 93% of companies in this industry have less than 250 employees and are classified as small and medium sized enterprises (SMEs). The development of technologies that are system specific rather than product specific means that they are integrated into existing plant and production lines, which is of clear benefit to an industry where SME investment is critical to change. This is the case for the Steam Infusion technologies demonstrated here, which reduce the heat consumption of manufacturing operations, and they have been used for several product categories including soups, sauces, and beverages. Industry 4.0 and Internet of Things (IOT) technology will begin to provide the added value of data collection at these integration points in processing, and this is demonstrated in the research reported here [6]. Advanced processing through heating technologies such as Steam Infusion have the potential to transform the food and beverage industry by reducing production downtime because they increase the speed of batch processing or provide continuous flow processing. This results in products reaching consumers in optimal quality through supply chains where there is greater ability to ensure production meets demand. This has important implications for meeting the Sustainable Development Goals (SDGs), and in the study presented here, it is SDG 9 (Industry, Innovation, and Infrastructure) and 12 (Responsible Consumption and Production) that are considered to be of the most importance [7].

Energy balance has an important impact on the preservation and packaging methods used, and improved efficiencies at these control points do have an impact on sustainability outcomes [8]. The integration of advanced processing heating technologies into these value streams is important because they are known to be a significant part of the carbon footprint for manufactured foods [9]. The systemic improvements of the Steam Infusion technologies reported here are evident in manufacturing schedules and show that they have the capacity to be effectively started and stopped or to have adjustments made to respond to demand changes and be more responsive to the distribution systems that deliver optimal product quality with less wastage [10]. This study considers such demand responsive production because it will be associated with the improved nutritional value of products, increased productivity, and reduced food waste through the whole food system because the viable shelf life is optimised. The distribution of brand value in the business ecosystem is critically important, and many SME’s supply branded products as ‘own label’ or ‘own brand’ for large named retailers so that brand value is effectively owned by the retailers. This relationship determines many routes for NPD and innovation where market agility is often determined by consumer trends identified by retailers who are focussed on the next priority consumer issue. The integration of heating technologies into development processes shows how agile technology is critical to providing responsive NPD and systemic change. When faster rates of processing decrease the heating-intensity experienced by a product through improved heat transfer, this provides exciting outcomes that are explored in this research. 

## 2. Materials and Methods

### 2.1. The Steam Infusion Process

The research reported was carried out at the National Centre for Food Manufacturing in a food factory demonstrator at Holbeach, UK. Steam Infusion is a technology that has been applied to novel methods of cooking liquid foods [11]. It injects culinary grade or ‘clean’ steam, into a liquid food inside a chamber where heating occurs. The typical chamber used in the food manufacturing sector for conventional cooking is called a kettle. It is jacketed in that the cooking or cooling process is separated from the heating or cooling liquids (e.g., steam or cooled water) by a steel walled container that holds a jacket of steam in the case of heating (Figure 1). The heat transfer in this conventional kettle, cooking occurs through a steel wall, and it is not in direct contact with the food materials. Typically, the kettle includes an internal agitator that scrapes the heated or cooled jacketed surface to reduce the build-up of an insulating layer, which in the case of steam heating, provides potential for Maillard reactions, resulting in burn-on to occur. The agitator also mixes the liquid food, which improves heat transfer, increasing heating efficiency. The Steam Infusion process is a direct contact heating process whereby steam is directly injected into the liquid food via the Vaction™ pump, which may be located within the kettle or within the recirculation line allowing the steam to be supplied from and returned to the kettle. There are no moving parts within the liquid food path, and it uniquely uses steam to simultaneously heat, mix, and pump the product. By changing the controlled steam pressure, the processing conditions are changed within the Vaction™ Pump to achieve the required product characteristics (Figure 2). The heating is achieved by the following process actions:Liquid food material is recirculated through the Steam Infusion Vaction™ pump, where steam is accelerated to a high velocity before entering into the liquid food stream via an annular nozzle (Figure 2). This process is conducted repeatedly with controllable temperature changes (∆t) between 1 °C and 30 °C, which are dependent on steam pressure. Through multiple cycles, an overall temperature gradient, typically up to 95 °C within the atmospheric kettle, is achieved. The product can be held at this temperature using an insulated covering so that the food materials are typically pasteurised and are cooked through to the required texture/consistency. Steam Infusion is designed to cook the liquid food extremely quickly and efficiently using steam that is under pressure and is in direct contact with food materials instead of through a heated jacketed kettle wall. It also provides options for using traditional kettle cooking to provide a ‘jacket and Steam Infusion’ production process. The trials for Steam Infusion were tested in this reported research using the steam infusion technologies developed by OAL (reported on-line through personal communication to W Martindale by OAL, Steam Infusion Test Centre, 2020). Profiling within the steam chamber can accelerate the velocity of the steam to 1000 m s^−1^ (3280 feet s^−1^), which is above the speed of sound. The steam passes into the mixing chamber through an annular nozzle disrupting the liquid food flow to form small droplets, referred to as the vapour phase. The momentum transfers from the steam to the food material and creates a partial vacuum of −0.7 barg (−10.1 psig) within the unit.As the steam condenses into the liquid food droplets, the pressure rises. This is referred to as the condensation shockwave, and it generates a pumping effect. The small droplets within the low-pressure vapour phase offer a significantly increased product surface area for the steam to condense into, typically resulting in a near instantaneous temperature gradient of 10–15 °C (50–59 °F).The very short residence times and partial vacuum within the unit prevent exposure to excessive temperatures. There are no hot contact surface areas/hot spots, and therefore, the Steam Infusion process prevents Maillard reactions and the resulting burn-on.An unrestricted Vaction™ Pump pumps at a rate of 50,000 kg/h (55 US t/h) in water at 20 °C (68 °F), and the turbulent mixing conditions in the low-pressure vapour area enhance the transfer of flavours. The Vaction™ Pump has an uninhibited bore of 47 mm (1.85 in), enabling particulates to freely pass through the unit with no damage.

### 2.2. Data Collection during Processing

A total of three process trials are reported in this research to determine the (1) processing energy and GHG balance, (2) processing time, and (3) volatile compounds in the food materials from typical steam jacket processing and Steam Infusion Vaction™ Pump heating. The time periods for cooking and processing were recorded for food material to a defined set temperature during each trial using a data logger for each cooking stage. Heat transfer was calculated during the cooking processes using the published specific heat capacities of the food materials so that an assessment for the efficiency of the steam infusion process could be made. The specific heat capacity (SHC) of product was calculated using published data for each ingredient and allocating these data to the recipe used, giving a mean SHC value [12]. Energy conversion factors for GHG emissions were obtained from the UK Government reported conversion factors (from the UK Department for Business Energy and Industrial Strategy 2020). The batch trials were as follows:(1)The processing energy and GHG balance trial processed batches of vegetable soup that had the following ingredients (specific recipe amounts are protected due to commercial sensitivity): butter, cream, milk, cabbage, mushroom, onion, potato, spinach, and tomato. As guidance, recipe typical amounts for a vegetable soup will be 35–45% *w*/*v* vegetables, 5% *w*/*v* butter, 5% *w*/*v* cream, 10% *w*/*v* milk, and processed to 100% volume with water. The soup was cooked using the Steam Infusion Vaction™ Pump and using a steam jacket vessel method separately to compare performance.(2)The processing time trial provided further temperature assessments and demonstrations of the Steam Infusion Vaction™ Pump data collection for water and a chocolate custard product. These were used as typical examples to show the diversity of products that can be processed with respect to time required to reach 90 C. These are reported to demonstrate both the data collection and the Steam Infusion Vaction™ Pump processing times achieved under trial conditions.(3)The volatile compounds in the product trials processed batches of a curry sauce that contained water (27% *w*/*v*), onion (7% *w*/*v*), tomato (15% *w*/*v*), rapeseed oil (5% *w*/*v*), tomato purée (10% *w*/*v*), ground spices (15% *w*/*v*), which comprised paprika, coriander, mustard, fenugreek leaf, and coriander leaf. The recipe used contained a modified maize starch (3% *w*/*v*), sugar (3% *w*/*v*), garlic purée (7% *w*/*v*), salt (3% *w*/*v*), and yeast extract powder (5% *w*/*v*).

Three trials report the energy balance, process time impact, and volatile analysis of typical steam jacketed kettle and Steam Infusion Vaction™ processing. This provides a platform for assessing the systemic impact of Steam Infusion Vaction™ Pump processing in the food and beverage industries. The steam jacketed kettle used for the trials also incorporated a Steam Infusion Vaction™ Pump to support the consistency of the comparisons made. The Steam Infusion Vaction™ Pump used generated steam from an oil fired boiler and used the GHG conversion factors provided by the UK Greenhouse Gas Conversion Factors for Company Reporting (Defra 2019) [13].

### 2.3. Analysis of Food Volatiles for Steam Infusion Cooking

Batches of a curry sauce produced by conventional steam jacket cooking and by Steam Infusion cooking were compared for different food headspace volatile compound composition using solid phase microextraction (SPME)-gas chromatography (GC)-mass spectrometry (MS) analysis (SPME GC-MS) analysis. The SPME GC-MS method sampled 3 g of each sauce and placed it in a 20 mL GC-MS headspace vial that was sealed with a cap. The samples were taken in triplicate and incubated for 1 h at 30 °C to establish vapour equilibrium and adsorption in the headspace with a Supelco DVB/CAR/PDMS SPME fibre. The SPME fibre was then manually loaded into the injection port of a Shimadzu GC-MS with the port temperature set at 270 °C and the injection port split ratio set at 25:1 with a Helium flow rate of 1 mL min^−1^. The gas chromatography used an Agilent J and W DB-1MS UI GC column that separates analytes based on their boiling point. The oven programme was 34 min in total duration with a 40 °C hold for 1 min, a ramp temperature at 10 °C min^−1^ to 270 °C, and a holding at 270 °C for 10 min followed by automated rapid cooling. The MS EI ion source temperature was 200 °C, the interface temperature was 270 °C, and the *m*/*z* scanning range was 35–500. The GC-MS procedure was repeated for three samples, the mean area of the chromatograph peaks was reported, and the separated volatile analytes were grouped into taste and aroma profiles according to published sources. 

The volatile compounds were categorised as the most probable ingredient and flavour component generated during the cooking process where compounds such as hexanals, β-pinene and euglenol, which are in a variety of herb and spice derived materials, can grouped according to chromatographic separation metrics in the Flavornet Database [14]. The Flavornet Database catalogues key odourants (KOs) that dominate natural products and are characterised using over 900 studies of KO separations. The headspace analysis did not provide information on non- or low-volatility compounds present in the body of the liquid product that could be responsible for taste (sweet, sour, bitter, salty, and umami) or flavour enhancing properties. A further consideration is that 30 °C is not the usual serving temperature for many of the products tested, and it was chosen to reduce oversaturation of the SPME fibre during the sampling time. 

## 3. Results

### 3.1. The Energy and CO_2_e Balance for Steam Infusion Processes

The food product tested in this research was a vegetable soup that contained butter, cream, milk, cabbage, mushroom, onion, potato, spinach, and tomato juice. This recipe has a specific heat capacity of 3.9 kJ kg^−1^ K^−1^ at a starting temperature of 12 °C and a final cooked temperature of 90 °C. The initial mass of the Steam Infusion cooked batch was 900 kg, and during the heating time of 14 min and 49 s, the addition of 104.4 kg of water as steam (through the heating process) occurred. In a typical cook–chill food soup/sauce manufacturing operation supplying retail and food service outlets, the number of batches cooked each year by each kettle is estimated at 2500, and the steam pressure is 3 bar g^−1^ for each batch in a conventional steam jacketed kettle cook. The number of conventional batch cooks taking place between cleaning in place (CIP) cleans is three, and this can be increased to six batches for Steam Infusion cooking processes (as Steam Infusion cooking avoids “burn on” to the side of the kettle caused by traditional jacket heating methods). Typically, each CIP cycle uses 500 litres of detergent/water, the amount of which is dependant on the degree of fluid recovery, and the cleaning process takes about 40 min to complete.

The heat energy consumed by 1004.4 kg of a typical food material (production batch) at the same point of use for Steam Infusion is 79.7 kWh and 96.4 kWh for conventional steam jacket heating. The steam introduced by the Steam Infusion Vaction™ Pump means 104.4 kg of water is added over the cooking time at a steam pressure of between 2 bar g^−1^ and 4.5 bar g^−1^ during each batch. Table 1 shows the expected efficiency improvements in terms of costs and greenhouse gas emissions for the Steam Infusion batch cooking over a period of one year using these operational data. The energy saving from each cooking and CIP process was GBP 1.13 for each cooked batch, an annual saving of GBP 2828 yr^−1^.

### 3.2. The Operational Efficiencies of Steam Infusion 

A conventional cooking process using a steam jacketed kettle will typically require 60 to 90 min to cook 400 kg of liquid soup and sauce food material and will hold at 90 °C. While batches can be greater than 1000 kg, the assessments conducted in this research were made for 400–500 kg batches. Using the Steam Infusion Vaction^TM^ Pump system reduces the cooking time to 9–15 min, and this is dependent of the specific heat capacity of the food materials. Figure 3A, shows the data log for the Steam Infusion heating of a 400 kg batch of water with the associated steam flow and total steam used. Figure 3B shows the Steam Infusion Vaction^TM^ Pump system data log for a 400 kg batch of chocolate sauce. The data logger records the resources used and the temperature history during the manufacture of a product batch. These data can be secured and used to confirm the provenance and quality attributes of the manufactured products. The graphics are derived from in situ reporting and the axes are for temperature (vertical) and processing time (horizontal). The water and chocolate custard batches are shown to demonstrate the diversity of products that can be processed. Energy saving is at the same point of use as per previous references. It does not consider feedback resulting in increased energy consumption at the boiler and assumes that the user is using condensate recovery systems where condensate traps are in effective condition.

### 3.3. Analysis of Volatiles in Steam Infusion 

Conventional steam jacket cooked and Steam Infusion cooked recipes were compared for their headspace volatile compound composition using SPME GC-MS analysis. Figure 4 shows the cumulative peak area totals for a GC-MS analysis of curry sauce batches that have been either steam jacket cooked or Steam Infusion cooked with all of the ingredients added directly into the cooking vessel; in addition, a steam infusion cooked batch with the spices added (entrained) from a hopper mid process when the batch product temperature was at 60 °C during the cook is also included. Volatile compounds were grouped according to general flavour/aroma notes to compare potential profile changes. Data are presented by plotting volatile peak areas and grouping them to the most probable ingredient and flavour component generated during the cooking process. 

## 4. Discussion

The Energy and CO_2_e Balance for Steam Infusion Processes: the operational efficiencies of Steam Infusion are demonstrated by the data shown in Table 1. Steam Infusion cooking reduces energy use per batch cook by 17.3%, and in a typical food production operation scenario, the additional reduction in the frequency of CIP related to use of Steam Infusion means 277.8 h of production time can be gained each year. An important sustainability metric reported for Steam Infusion is the energy reduction, and the faster rate of production means that there is a GHG emission reduction of 8.7 CO_2_e annually. The Vaction Pump^TM^ heating energy at the point of use was 79.7 kWh with a steam addition at 4.5 bar g^−1^ and 2 bar g^−1^ at point of use of 104.4 kg, recording a final heated mass of 1004.4 kg after 14 min 49 s. Steam jacket heating energy at the point of use was 96.4 kWh with a steam addition at 3 bar g−1 at the point of use of 162.8 kg, recording a final heated mass of 1004.4 kg and resulting in 277.8 additional processing hours over a typical year production cycle. This resulted in an annual energy saving of 38.4 MWh, equivalent to 8.7 tonnes of GHG (CO_2_e) at 227 kg CO_2_e. MWh^−1^. This significantly increases site output capacity and thereby extends the amount of time during business growth before the manufacturer has to make financially and environmentally costly extensions to the production site or build an entirely new factory. Figure 3A,B, show how the processing times are reduced by using the Steam Infusion Vaction^TM^ Pump system. While the downtime and CIP savings made using the technology are crucial to the immediate ROI, it is the emergence of IOT within the Industry 4.0 system that offers opportunities for recording data during processing that can be connected to quality outcomes and supply chain inventory. There are already established mechanisms for determining the provenance of wine and oil products using volatile and sensory analysis, and the use of specific processing technologies such as Steam Infusion can provide a further means of establishing traceability for provenance and quality [15]. The issue of traceability is also going to be of importance in determining the environmental impact of imported products. The data control methods for Industry 4.0 are the most incisive when they are established in the processing sector because this is where ingredients are sourced, products are manufactured, and delivered. In many respects, it is the data nerve centre of the food supply chain. This is likely to be made possible through the embedding of IOT applications in the processing and manufacturing arena. The demonstrations reported here do begin that process because ROI is proven, and data collection is necessary for product time–temperature history.

Analysis of Volatiles in Steam Infusion Cooking: reduced thermal process time and minimal exposure of a product to hot surfaces in the Steam Infusion cooked batches results in improved retention and generation of flavour and aroma volatiles. Figure 4 shows that these flavours are retained more in the Steam Infusion cooking process, and when the flavouring ingredients are added during the cooking process (‘entrained’), flavours can be further enhanced. The Steam Infusion process is also likely to be of benefit to the processing of dairy based products that are sensitive to well-developed Maillard products and overall thermal exposure, such simple white sauces (béchamel, parsley sauce) and custards. A benefit of traditional steam injectors developed for use in jacketed kettles is that the use of steam injection helps prevent burn-on to the jacket sides (when compared to cooking via a jacket only method). Dairy based products are susceptible to casein precipitation (protein denaturation) during increased temperature spikes in the cook, and therefore, injectors reduce this because the heat is quickly distributed in the kettle, giving more control of the heating process (personal communication to W Martindale OAL 2020, see Figure 3A,B) [16]. These advanced processes are important because they improve the consumer food experience, and this not only enhances brand value but also results in more responsible consumption (SDG 9) because if a food is preferred, it has a reduced risk of being wasted [8]. 

A limitation of traditional steam injectors is that of heat transfer rate. Operation at too high a pressure may cause steam bubbles to pass through liquid food material that is being heated and may simply break the surface of the fluid and escape to the atmosphere instead of condensing into the bulk contents. Hence, some of the heat contained in the steam will be lost to the atmosphere, and actual heat transfer to the food material will be reduced. This loss can be controlled by the restriction of the operating steam pressure and the use of multiple injection points, but this increases the complexity of the steam distribution pipework and increases potential points of food material hold-up [17]. It is this aspect that differentiates Steam Infusion processing from the traditional steam injector technology. The steam condenses into the liquid within the Vaction™ Pump, which is aided by the large surface area of the small disrupted liquid food material droplets, the low-pressure vapour phase, rather than within the bulk contents of the kettle. This allows for processing across a range of operating pressures, providing for the consistent control of the physical characteristics of the liquid food material as well as effective heat transfer.

The Steam Infusion cooking process has been shown to provide advantages in milk processing, where results show milder flavour profiles with more volatiles coming from the ingredients [11]. The application of Steam Infusion to post process food materials for antinutrient inactivation has also been successfully tested for soy products [18]. The research reported in this study extends the potential of Steam Infusion flavour retention, being applied to recipes with herbs, spices, fruits, and chocolate. Figure 4 shows data that demonstrate that herb and spice volatiles are increased by Steam Infusion production. This suggests less flavouring is required to achieve the same impact, and future investigations will identify this and the impact of Steam Infusion associated with meat and dairy products, where there is often a requirement to develop flavours. Changes to the carbon footprint are dependent of energy supply to the factory, and the Steam Infusion Vaction^™^ Pump used steam generated from an oil fired boiler and used the GHG conversion factors provided by the UK Greenhouse Gas Conversion Factors for Company Reporting (Defra 2019). [13].

Figure 4 shows the Steam Infusion cook with the spices added (entrained) from a hopper mid process at 60 °C during the cook. These trials showed that by entraining the spices and reducing the time that they were in the process increases the retention of volatiles, raising the possibility that reformulation opportunities exist using Steam Infusion technologies. This may result in salt and sugar reduction opportunities because there is evidence that when flavouring ingredients are mixed uniformly, salt and sugar reduction are possible [19]. The impact of starch gelatinisation in the Steam Infusion process will be defined in future research reports. There is no difference between Steam Infusion and conventional kettle cooking with respect to the stability of products and syneresis (separation of starch, aqueous and oil phases), but there are indications that starch gelatinisation is more efficient in Steam Infusion cooking, and this may allow for the reduction of fat content in foods [17].

The sustainability outlook and step change opportunity for manufacturing: the metrics used to measure both the nutrition and sustainability of foods have been used in this investigation to demonstrate the further benefits of using Steam Infusion cooking [20]. Research shows the GHG emission per calorie or mass of protein associated with soup, sauce, and preserved vegetable categories is greater than other food product categories [21]. The use of life cycle assessment (LCA) data such as the global warming potential are routinely reported and tested for many food categories, where the methods used to do this identify food categories in which there are specific barriers to producing food products with increased sustainability attributes [22]. This is the case for soup, sauce, and preserved vegetable categories that have high nutritional density but also have high GHG emissions per calorie. These products are therefore an opportunity for GHG reduction interventions such as the use of Steam Infusion cooking. The reduction in energy used to process these categories using Steam Infusion provides GHG reduction and nutritionally favourable outcomes. The innovative heating technology demonstrated in this study will reduce the GHG emission footprint of these nutrient dense but low calorie or low protein product categories and overcome the sustainability risks associated with them. The major risk is that high nutritional density products will be viewed as high carbon footprint products by consumers.

The Steam Infusion technologies used in these reported trials are modular, and they can be used when demand is required so that production becomes responsive to consumption trends. This results in efficiency improvements in factories, and the resulting productivity increases are being demonstrated through the global implementation of Steam Infusion (see, OAL 2020). These have the potential to influence NPD through responsiveness in the reformulation of products such as in the case of sugar and salt reduction. The value of Steam Infusion in creating a modular and responsive food system is demonstrated here, and the value of it to reformulation will be developed in future research reporting. The faster consumer goods can be packed and distributed following heat treatment, the greater proportion of the product shelf life can be used in the supply chain, and this consequently reduces the risk of wasting products. The disruptive effect of these technologies in this advanced processing arena is enhanced by their modulisation because they can be installed and activated on existing conventional production lines, known as “retro-fitting”, in a very short time. The technologies reported here enhance the value of ROI opportunities in the food and beverage sector to deliver innovation and improved industrial infrastructure (included in Sustainable Development Goal 9). They are immediately available to be used by manufacturers, and this research quantifies their operational and related sustainability impacts, providing new insights into how NPD strategies can be transformed. The development of multi-ingredient products such as soups, sauces, and preserved vegetable products becomes exceptionally important because they are at the core of popular ‘ready to prepare’ or ‘ready to eat’ meal formats. As such, Steam Infusion is an excellent example of advanced food processing technology that is systemically transforming the manufacturing, retail, and consumption functions of food supply.

## 5. Conclusions

Steam Infusion processing offers innovative processing solutions, and this research demonstrates the potential for it to reduce the greenhouse gas emissions of production. This is an important intervention if the food and beverage industry is to follow current route maps for carbon-zero production that are not only reliant on off-setting greenhouse gas emissions using mechanisms external to the business enterprise and manufacturer. Steam Infusion has also been demonstrated to provide important quality outcomes including the reduced risk of acrylamide forming compounds and the improved bioavailability of nutrients (personal communication from C.B. OAL ). These outcomes are characterised and reported by the OAL , and this research reports a reduction in energy used on a single Vaction Pump^TM^ production line of 17.3% and 277.8 production hours compared to direct steam kettle heating production. In addition to providing greater operational agility because of a reduction in production time, there is a reduction of 8.7 tonnes CO_2_e.

## Figures and Tables

**Figure 1 foods-10-01763-f001:**
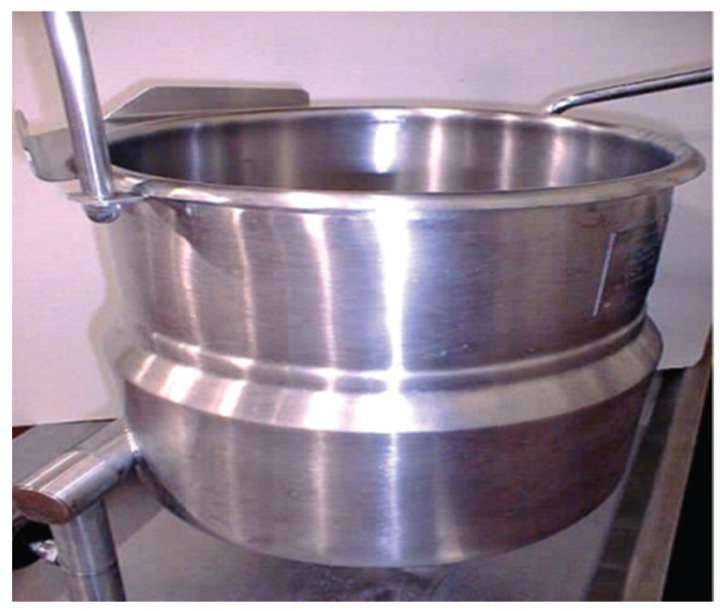
A conventional steam jacketed kettle used to heat batches of food materials. The kettle chamber is typically 1.5 m in diameter with a 1.5 m depth.

**Figure 2 foods-10-01763-f002:**
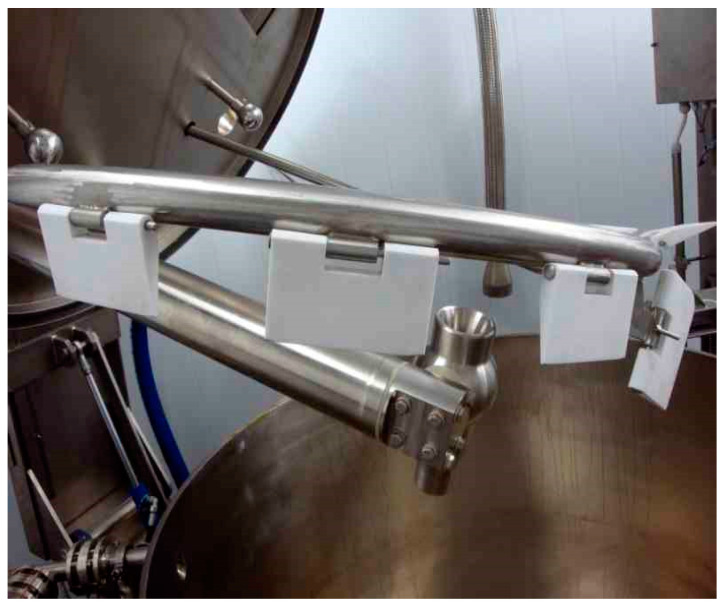
The OAL Steam Infusion Vaction™ pump mounted in a cooking kettle with a scraped surface agitator. Image provided by OAL.

**Figure 3 foods-10-01763-f003:**
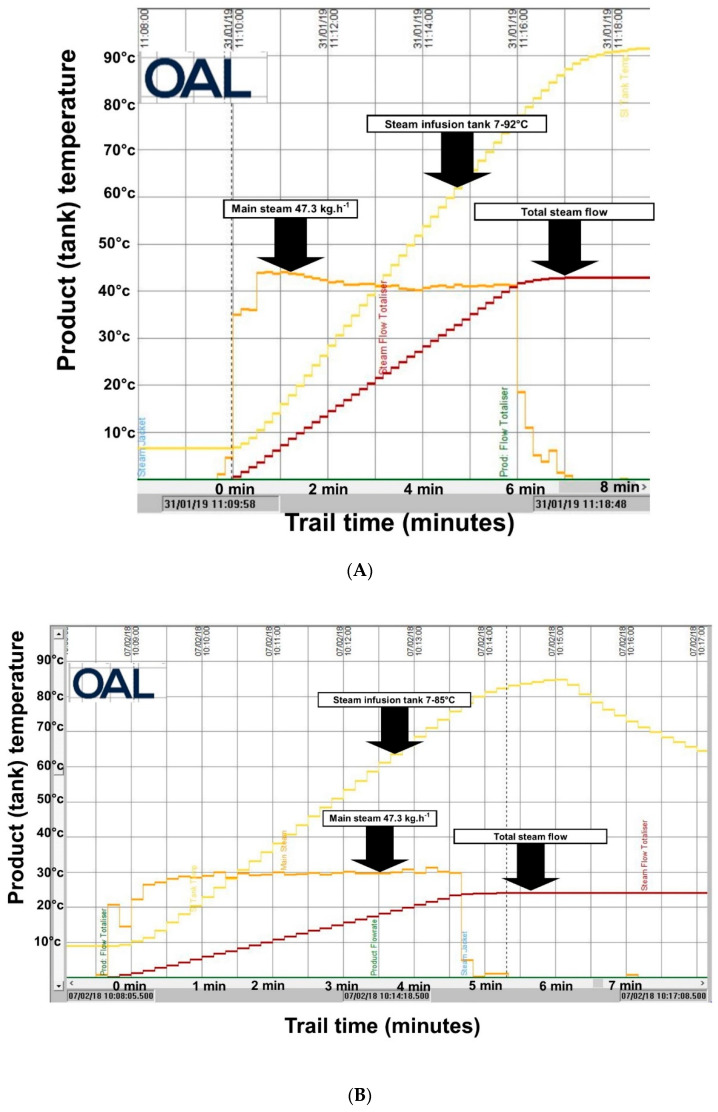
(**A**) The Steam Infusion Vaction^TM^ Pump system data for the heating of 400 kg water. The temperature in the Steam Infusion heated kettle starts at 7 °C and finishes at 92 °C over the duration of the cooking time of 8 min. (**B**) The Steam Infusion Vaction^TM^ Pump system data for the heating of 400 kg chocolate custard. The temperature in the Steam Infusion heated kettle starts at 9 °C and finishes at 85 °C over the duration of the cooking time of 6.5 min.

**Figure 4 foods-10-01763-f004:**
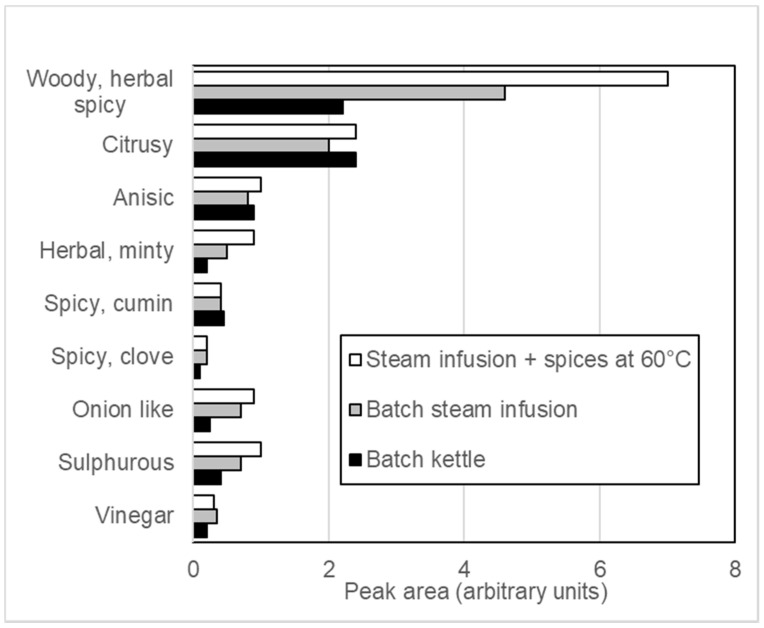
The cumulative peak area totals for a GC-MS analysis of curry sauce batches that have been cooked by either conventional steam jacket (“Batch Kettle”) or Steam Infusion with all of the ingredients batched in the kettle; a Steam Infusion cook with the spices added (entrained) from a hopper mid process at 60 °C batch temperature during the cook is also included.

**Table 1 foods-10-01763-t001:** The efficiency of Steam Infusion cooking compared to conventional steam jacketed vessel cooking.

Steam Infusion Operational Efficiency	
Energy reduction at point of use compared to steam jacket vessel cooking (% reduction)	17.3
Production time saved by Steam Infusion cooking method related CIP reduction (cleaning time hours reduced per year)	277.8
GHG reduction due to decreased energy use (CO_2_e reduction per year)	8.7

## Data Availability

Data that further support the reported results are held under an NDA between the University of Lincoln and OAL and are available upon request and by permission of the project partners. This research and supplementary data are reported by OAL at https://www.oalgroup.com/ (accessed on 27 July 2021).

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
