# Peer review of "Transformational Steam Infusion Processing for Resilient and Sustainable Food Manufacturing Businesses"

_foods, 2021, doi:10.3390/foods10081763_

Round 1
Reviewer 1 Report
The authors describe a new process for processing food. The steam infusion processing process shows interesting results and energy savings in comparison to conventional methods. The presentation of the process is okay and the experimental set-up is adequately described. The evaluation of the results and the presentation of the data are too abbreviated, so that it is difficult for the reader to understand the results and data sufficiently. Improvements must be made here.
In addition, I ask to improve the following points.
- The introduction is very broad and focuses on the change in the industry by introducing ne technologies. I am missing the literature review about energy efficiency in food manufacturing and how this new approach goes beyond the measurements in other papers. Please reduce the general introduction and put your work more in context to other research.
- Why use a recipee that is proprietary due to commercial sensitivity? Using a publicly available recipee increases the reproducibility of the results.
- After the first figures the formatting needs to be improved.
- In section 3.1 the process data is well described. For the reader, it would be benefical if a table with the comparison of the new and the conventional process is provided.
- Figure 3A and B have no description of the axes and are of poor quality. Please improve the two figures.
Why do the figures 3 not show the results for the descriped products in section 3.1? - Line 331: Please show the calculations for the presented GHG emission reduction. How much steam is saved, how much water? Which GHG emissions factors are you considering? How is the steam provided for your comparison?
Author Response
We thank you for your robust review of our paper and we address each of your comments below where we have amended figures and text we refer to these in the responses below. You review and comments have made the initial draft submitted much more robust and significant. We thank you for this.
Response to academic editor comments
Good ms however, the comparison is not fair and could be a bias towards this particular technology. Please revise to make it more objective.
We have revised the text to accommodate this top-line issue. The issue of bias and unfairness is addressed in that we may not have made it clear in out initial submission that the Vacation Pump device provides a different approach in that the steam infusion is not transferred through the material in the kettle/tank. This is the traditional means of heating materials. The Vacation Pump device acts so that the steam disruption occurs within the device and does not transfer the heat through the kettle. Typically steam infusion into the kettle will transfer heat into the kettle. This new approach occurs within the device and increasing steam pressure increases the velocity at which food materials passes through the device. It does not introduce more steam into the kettle (increasing the volume of liquid in the kettle).
It is therefore a different approach to heating liquids that results in more directed heating of the foods with less steam introduced into the kettle and heat lost to the atmosphere through the top of the kettle. Any bias observed by the reviewers is not intentional, the intention is to describe the novel heating mechanism that occurs in the device/Vacation Pump. Most importantly the focus of this paper is to highlight the energy reduction and GHG reduction because of the faster processing times and reduction of steam loss in heating and processing the foods. We have kept the description of product to a minimum so as not to introduce commercial bias and keep our focus on process efficiency.
We have edited the text to reflect this focus, describe the novel process and I hope we have explained any potential inference of bias.
- Compared steam infusion with scraped surface mixer to a static st4eam jacketed kettle. Efficiencies in a steam jacketed (indirect) kettle can be enhanced significantly by mixing. Why not use the same kettle with infusion vs. indirect heating? It is not a fair comparison as they did here. They will have to note and explain this.
We address this in Lines 130-140 in the resubmitted MS. The text now refers to scrapers and mixing. The study was not intended to be a comparison is was to report the efficiency of the Vaction Pump device with regard to improved processing time and energy reduction. These were directly compared to steam jacket heating. See Table 1, lines 285-290
2. The abstract does not contain meaningful info. An abstract is a concise summary of the ms, with objectives, summary of methods and results
We have edited the abstract to reflect the comments made with regard to comparison of steam infusion technologies and general comments so that it is more incisive.
3. Some terms like "reduction in heat transfer energy" is not correct. You have reduction in heat consumption...
We have corrected this.
4. The ms needs to be reviewed by experts in the technology/food processing. This technology is not new, the pump concept may be new and thus why not compare to other existing steam infusion technologies?
The pump concept and associated device is new and unique. We have not emphasised the detail regarding this because it would introduce commercial bias. We are seeking to link improved efficiency by using such new technologies to quality and sustainability attributes.
Reviewer 1 responses
The authors describe a new process for processing food. The steam infusion processing process shows interesting results and energy savings in comparison to conventional methods. The presentation of the process is okay and the experimental set-up is adequately described. The evaluation of the results and the presentation of the data are too abbreviated, so that it is difficult for the reader to understand the results and data sufficiently. Improvements must be made here.
In addition, I ask to improve the following points.
- The introduction is very broad and focuses on the change in the industry by introducing ne technologies. I am missing the literature review about energy efficiency in food manufacturing and how this new approach goes beyond the measurements in other papers. Please reduce the general introduction and put your work more in context to other research.
Thank you for this, we have identified where there are sentences that do not provide the detail required and deleted them. However, we do feel that we need to emphasise the connection between universal processes such as heating in factories that are the same energy consuming processes for most food businesses and the need to retrofit more efficient processing. We feel the current Introduction does this and have emphasised this point using your helpful review. We clearly did not emphasise this fully enough. It is an important point- heat processes and treatments are expensive, reducing them is a system need and being able to retrofit at relatively lower investment means it is open to SME use. This is transformative in our view and worthy of note in the introduction. Our focus is not the steam infusion device, it is the process and benefits conferred when the technology is integrated. - Why use a recipee that is proprietary due to commercial sensitivity? Using a publicly available recipee increases the reproducibility of the results.
You are right, we have included guidance values, lines, 210-214. - After the first figures the formatting needs to be improved.
We have reviewed and edited - In section 3.1 the process data is well described. For the reader, it would be benefical if a table with the comparison of the new and the conventional process is provided.
We have reviewed this and would introduce a flow diagram or table but think this would add to the length of the MS and repeat what we state in the text, we have considered this in writing the MS but decided the detail in the text was required and the best way to do this as highlighted by the reviewer- ‘it is well described’. - Figure 3A and B have no description of the axes and are of poor quality. Please improve the two figures.
We have re-drawn the graphs, these are in situ factory outputs and we are constrained by the bespoke software, However, these are calibrated and relevant to our study
Why do the figures 3 not show the results for the descriped products in section 3.1?
We wish to show the versatility of the processing and present data in Table for the soup and Figures for water and a chocolate sauce. We now emphasise this in the text- we hope this can accommodate the reviewers thoughts. We have defined this more robustly improving the MS, thank you. - Line 331: Please show the calculations for the presented GHG emission reduction. How much steam is saved, how much water? Which GHG emissions factors are you considering? How is the steam provided for your comparison?
We show direct measurements from our raw data here, thank you for this comment, we have strengthened the text in response to the review.
Reviewer 2 responses
Congratulations for your interesting work.
But it can be improved:
- Some results can be added to the abstract, for instance results of table 1.
- Introduction can be shortened
- Conclusion can be improved- add some results
- The rest looks fine to me
We have addressed these helpful comments and have included the results in our abstract and conclusion. We have edited the introduction with reflection of both reviewers, the review has strengthened the introduction. Thank you for your help
Please see attachment

Reviewer 2 Report
Congratulations for your interesting work.
But it can be improved:
- Some results can be added to the abstract, for instance results of table 1.
- Introduction can be shortened
- Conclusion can be improved- add some results
- The rest looks fine to me
Author Response
We thank you for your robust review of our paper and we address each of your comments below where we have amended figures and text we refer to these in the responses below. You review and comments have made the initial draft submitted much more robust and significant. We thank you for this.
Response to academic editor comments
Good ms however, the comparison is not fair and could be a bias towards this particular technology. Please revise to make it more objective.
We have revised the text to accommodate this top-line issue. The issue of bias and unfairness is addressed in that we may not have made it clear in out initial submission that the Vacation Pump device provides a different approach in that the steam infusion is not transferred through the material in the kettle/tank. This is the traditional means of heating materials. The Vacation Pump device acts so that the steam disruption occurs within the device and does not transfer the heat through the kettle. Typically steam infusion into the kettle will transfer heat into the kettle. This new approach occurs within the device and increasing steam pressure increases the velocity at which food materials passes through the device. It does not introduce more steam into the kettle (increasing the volume of liquid in the kettle).
It is therefore a different approach to heating liquids that results in more directed heating of the foods with less steam introduced into the kettle and heat lost to the atmosphere through the top of the kettle. Any bias observed by the reviewers is not intentional, the intention is to describe the novel heating mechanism that occurs in the device/Vacation Pump. Most importantly the focus of this paper is to highlight the energy reduction and GHG reduction because of the faster processing times and reduction of steam loss in heating and processing the foods. We have kept the description of product to a minimum so as not to introduce commercial bias and keep our focus on process efficiency.
We have edited the text to reflect this focus, describe the novel process and I hope we have explained any potential inference of bias.
- Compared steam infusion with scraped surface mixer to a static st4eam jacketed kettle. Efficiencies in a steam jacketed (indirect) kettle can be enhanced significantly by mixing. Why not use the same kettle with infusion vs. indirect heating? It is not a fair comparison as they did here. They will have to note and explain this.
We address this in Lines 130-140 in the resubmitted MS. The text now refers to scrapers and mixing. The study was not intended to be a comparison is was to report the efficiency of the Vaction Pump device with regard to improved processing time and energy reduction. These were directly compared to steam jacket heating. See Table 1, lines 285-290
2. The abstract does not contain meaningful info. An abstract is a concise summary of the ms, with objectives, summary of methods and results
We have edited the abstract to reflect the comments made with regard to comparison of steam infusion technologies and general comments so that it is more incisive.
3. Some terms like "reduction in heat transfer energy" is not correct. You have reduction in heat consumption...
We have corrected this.
4. The ms needs to be reviewed by experts in the technology/food processing. This technology is not new, the pump concept may be new and thus why not compare to other existing steam infusion technologies?
The pump concept and associated device is new and unique. We have not emphasised the detail regarding this because it would introduce commercial bias. We are seeking to link improved efficiency by using such new technologies to quality and sustainability attributes.
Reviewer 2 responses
Congratulations for your interesting work.
But it can be improved:
- Some results can be added to the abstract, for instance results of table 1.
- Introduction can be shortened
- Conclusion can be improved- add some results
- The rest looks fine to me
We have addressed these helpful comments and have included the results in our abstract and conclusion. We have edited the introduction with reflection of both reviewers, the review has strengthened the introduction. Thank you for your help
Please see attachment

Round 2
Reviewer 1 Report
The raised questions and concerns have been answered and improved in the new submission.
Two points need to be corrected.
- Line 229 - Please add the source for the ghg emissions factors to the reference list
- Line 444 - 445 - Please explain briefly your used ghg factor for steam. Is the steam provided by a gas or electrical boiler? If electrical, which grid emission factor are you using?
Author Response
Review 2 Responses from authors
Thank you for further comments and your observations have provided a more robust article. We thank you for that.
We address your review and comments below.
We have included four additional references in the text as follows. We are convinced that the use of steam infusion in academic literature is not fully represented. The use of localised steam infusion with the pump technology presented in this manuscript provide s a new approach that reduces energy and improves manufacturing efficiency. This is a change of prior reported techniques, the literature supporting this previously is scant. This paper presents one of the first if not the first to do this. This is why the literature is limited. The is an opportunity to change this here with this paper as a first.
- Line 463: Fayle, S.E.; Gerrard, J.A. The maillard reaction; Royal Society of Chemistry, 2002; Vol. 5; ISBN 0854045813.
- Line 451: Chen, S.H.; Li, X.-F.; Shih, P.-T.; Pai, S.-M. Preparation of thermally stable and digestive enzyme resistant flour directly from Japonica broken rice by combination of steam infusion, enzymatic debranching and heat moisture treatment. Food Hydrocoll. 2020, 108, 106022.
- Line 408: Rauh, V.M.; Sundgren, A.; Bakman, M.; Ipsen, R.; Paulsson, M.; Larsen, L.B.; Hammershøj, M. Plasmin activity as a possible cause for age gelation in UHT milk produced by direct steam infusion. Int. Dairy J. 2014, 38, 199–207.
- Line 423 and 456: Bowser, T.J. Steam basics for food processors. 2006. Oklahoma: Oklahoma Cooperative Extension Service. Available from http://osufacts.okstate.edu/docushare/dsweb/Get/Document-3042/FAPC-142web.pdf [accessed on 21st January 2021].
See line 233
The Steam Infusion Vaction™ Pump used generated steam steam from an electric boiler and used the GHG conversion factors provided by the UK Greenhouse Gas Conversion Factors for Company Reporting (Defra 2019). (insert reference for Defra 2021 UK Greenhouse Gas Conversion Factors for company Reporting 2019. See, https://www.gov.uk/government/collections/government-conversion-factors-for-company-reporting accessed 12th July 2021).
See Line 451
Changes to the carbon footprint are dependent of energy supply to the factory and the Steam Infusion Vaction™ Pump used generated steam from an electric boiler and used the GHG conversion factors provided by the UK Greenhouse Gas Conversion Factors for Company Reporting (Defra 2019). (insert reference for Defra 2021 UK Greenhouse Gas Conversion Factors for company Reporting 2019. See, https://www.gov.uk/government/collections/government-conversion-factors-for-company-reporting accessed 12th July 2021).
